Overview of current state of research on the application of artificial intelligence techniques for COVID-19

Kumar Vijay 1
http://orcid.org/0000-0001-6475-4491 Singh Dilbag 2
Kaur Manjit 2
http://orcid.org/0000-0001-9990-1084 Damaševičius Robertas 3 4 robertas.damasevicius@vdu.lt
1 Computer Science and Engineering Department, National Institute of Technology , Hamirpur, Himachal Pradesh , India
2 School of Engineering and Applied Sciences, Bennett University , Greater Noida , India
3 Faculty of Applied Mathematics, Silesian University of Technology , Gliwice , Poland
4 Department of Applied Informatics, Vytautas Magnus University , Kaunas , Lithuania
Zubiaga Arkaitz
Electronic publication date: 2021 May 26
Publication date: 2021
Volume: 7
Electronic Location ID: e564
Received 2020 Sep 17; Accepted 2021 May 5
Copyright: © 2021 Kumar et al.
Copyright year: 2021
Copyright holder: Kumar et al.
License: This is an open access article distributed under the terms of the Creative Commons Attribution License, which permits unrestricted use, distribution, reproduction and adaptation in any medium and for any purpose provided that it is properly attributed. For attribution, the original author(s), title, publication source (PeerJ Computer Science) and either DOI or URL of the article must be cited.
License URL: https://creativecommons.org/licenses/by/4.0/

Keywords: Artificial intelligence, Disease prediction, Diagnosis, Covid-19

Funding: The authors received no funding for this work.

==============================
Background

Until now, there are still a limited number of resources available to predict and diagnose COVID-19 disease. The design of novel drug-drug interaction for COVID-19 patients is an open area of research. Also, the development of the COVID-19 rapid testing kits is still a challenging task.

Methodology

This review focuses on two prime challenges caused by urgent needs to effectively address the challenges of the COVID-19 pandemic, i.e., the development of COVID-19 classification tools and drug discovery models for COVID-19 infected patients with the help of artificial intelligence (AI) based techniques such as machine learning and deep learning models.

Results

In this paper, various AI-based techniques are studied and evaluated by the means of applying these techniques for the prediction and diagnosis of COVID-19 disease. This study provides recommendations for future research and facilitates knowledge collection and formation on the application of the AI techniques for dealing with the COVID-19 epidemic and its consequences.

Conclusions

The AI techniques can be an effective tool to tackle the epidemic caused by COVID-19. These may be utilized in four main fields such as prediction, diagnosis, drug design, and analyzing social implications for COVID-19 infected patients.

Introduction

The novel coronavirus has been reported in Wuhan (China) in December 2019. Wuhan became the epicenter of coronavirus (Wu et al., 2020). Coronavirus infected 138,987,378 persons and 2,988,860 deaths in 210 countries as on 15 April, 2021 (Huang et al., 2020; Zou, Shu & Feng, 2020). World Health Organization (WHO) entitled the disease caused by coronavirus as COVID-19 and declared it an epidemic in February 2020 (Ayeningbara, 2020). The virus, also known as SARS-CoV-2, is a novel and evolving virus. The treatment of SARS-CoV-2 is based on the symptoms present in the patient. The most common and specific symptoms are fever and cough with some other non-specific symptoms such as fatigue, headache, and dyspnea (Shi et al., 2020). Table S1 shows the contribution of these symptoms in infected persons (Jiang et al., 2020). The common transmission methods are human contact and respiratory droplets. The COVID infection depends upon age, preexisting health conditions, hygiene and social habits, location, and frequency of person interactions (Rosenberg, Syed & Rezaie, 2020).

Table S2 shows the estimation of the severity of COVID-19 disease infected peoples (Regmi & Lwin, 2020; Verity et al., 2020). The risk associated with infection is broadly classified into three categories namely infection risk, severity risk, and outcome risk (Lalmuanawma, Hussain & Chhakchhuak, 2020). The infection risk is associated with a specific group/person having COVID-19. The person/group having severe symptoms of COVID-19 and require intensive care and hospitalization is known as severe risk. If the treatment is not effective towards the infected person/group then there is less possibility to recover or die. Self-isolation and social distancing are the most effective strategy to alleviate this epidemic. Isolation and house quarantine are core strategies to alleviate the infectious disease and reduce the transmission via diminishing the contact of those that are infected (Alimadadi et al., 2020). Most governments have imposed lockdown to save lives. However, the economy of every country was greatly affected by the lockdown process. The Organization of Economic Cooperation and Development declared that the growth rate may be slow down by 2.4% (Rao, Agarwal & Batura, 2020).

The main problems associated with coronavirus’s pandemic can be resolved through Artificial Intelligence (AI) (Allam, 2020). AI has the potential to screen the population and predicting the risk of infection. The prediction process utilizes information such as how much time a person spends in a highly infected area and how many persons are infected in that region. AI developed a spatial prediction model based on this information and envisage the infection transmission (Liu et al., 2020). AI-based prediction model uses information about the infected person through the symptoms. Coronavirus infected person does not show the symptoms in most of the cases. Due to this, it is very difficult to detect an infected person. The same thing has been reported in Wuhan city. In Wuhan, 50% of the infected persons are asymptomatic carriers (Bullock et al., 2020). The exhaustive testing of coronavirus is required to develop a better predictive model. The same scenario has been implemented in South Korea to prevent the spread of virus infection. The exhaustive testing produces a large amount of data about the infected and non-infected person. Based on these data, AI can be used to suppress the spread of infection, development of vaccines, diagnosis, and social-economic impact (Singhal, 2020; Phan et al., 2020). Recently, most of the AI researchers are working on the above-mentioned areas. The number of preprints available on the Internet is a witness of this work (Majumder & Mandl, 2020). Recently, many medical image processing techniques based upon chest CT and chest X-ray images have been considered. Also, meta-heuristics techniques can be useful to diagnose COVID-19 patients (Senthilraja, 2021; Kumar, Kaur & Singh, 2021).

By the end of November 2020, more than 75,000 scholarly articles were published and indexed on Pubmed on COVID-19 (Chen, Allot & Lu, 2020). However, these articles did not address in-depth the key issues in applying computational intelligence to combating the COVID-19 pandemic. Thus, it is time to discuss and summarize studies related to artificial intelligence from such a large number of articles. Considering the above observations, now it is the time to systematically categorize and review the current progress of research on artificial intelligence. Accordingly, this survey aims to assemble and summarize the highlights of the latest developments and insights in applying artificial intelligence approaches, such as machine learning, and deep learning, to practical applications used to fight against COVID-19.

The principal contributions of this review paper are as follows:The impact of artificial intelligence techniques is assessed on early cautionary and vigilant systems.

COVID-19 datasets and visualization techniques are discussed with their applications.

AI-based diagnosis and treatment of COVID patients are assessed.

The impact of AI on drug discovery and design is evaluated.

The effect of COVID-19 is evaluated on social and economic aspects.

The remaining paper is organized as follows. “Research Methodology” presents a description of our methodology used. “Coronavirus Overview” presents a short description of coronavirus. The data gathering tools and techniques are discussed in “Data Gathering Systems for COVID-19”. “Prediction Models” presents the prediction models. The diagnosis of infected patients is presented in “Screening Techniques for COVID-19”. “Diagnosis of COVID-19” describes the treatments of COVID patients. The drug discovery and design of vaccines are presented in “Drug Design”. The social and economic impacts are discussed in “Social Implications”. Future research directions are discussed in “Future Research Directions”. The concluding remarks are drawn in “Findings, Lessons and Recommendations”.

Research Methodology

In this section, the surveys related to application of artificial intelligence towards COVID-19 are discussed followed by the research paper selection methodology used in this paper.

Related surveys

Since the start of the COVID-19 pandemic, many research paper was published (just Scopus has already indexed over 50,000 papers alone). Consequently, several surveys and systematic review studies have tried to systemize and summarize the state of research and knowledge in this emerging research sub-field, including on the use of artificial intelligence methods. Related survey papers are summarized in Table S3 and discussed in more detail below.

Albahri et al. (2020) reviewed the state-of-the-art techniques for coronavirus prediction algorithms based on data mining and ML assessment. They used Preferred Reporting Items for Systematic Reviews and Meta-Analyses (PRISMA) as a methodological guideline. The main focus of the survey study was on the development of different AI and ML applications, systems, algorithms, methods and techniques. However, only eight articles were fully evaluated and included in this review, which outlined the insufficiency of research in this important area.

Lalmuanawma, Hussain & Chhakchhuak (2020) aimed to review the role of AI and ML as one significant method in the arena of screening, predicting, forecasting, contact tracing, and drug development for SARS-CoV-2. The study concluded that the use of modern technology with AI and ML dramatically improves the screening, prediction, contact tracing, forecasting, and drug/vaccine development although they also noted the lack of deployment of AI models to show their real-world operation.

Ozsahin et al. (2020) analyzed the use artificial intelligence (AI) techniques to diagnose COVID-19 with chest computed tomography (CT). Their study included 30 articles from ArXiv, MedRxiv, and Google Scholar identified using the selective assessment method.

Pham et al. (2020) present an overview of AI and big data, then identify the applications aimed at fighting against COVID-19, next highlight challenges and issues associated with state-of-the-art solutions, and finally come up with recommendations for the communications to effectively control the COVID-19 situation. They based their study on the selected assessment of peer-reviewed papers and preprints from IEEE Xplore, Nature, ScienceDirect, Wiley, arXiv, medRxiv, and bioRxiv.

Rasheed et al. (2020) presented the collation of the current state-of-the-art technological approaches applied to the context of COVID-19, while covering multiple disciplines and research perspectives.

Tseng et al. (2020) focused on categorizing and reviewing the current progress of computational intelligence for fighting COVID-19, which additionally to machine learning and neural networks also discuss fuzzy logic, probabilistic and evolutionary computation based methods. Tayarani (2021) presented the applications of artificial intelligence techniques in COVID-19. They discussed the machine learning techniques for prediction and treatment of infected persons.

However, since the body of knowledge on COVID-19 related research problems is rapidly updated and supplemented, there is a need to provide a new survey of papers to summarize the most recent state-of-the-art in the application of AI techniques for in the COVID-19 related research fields.

Survey methodology

This survey capitalizes on previous literature to describe approaches for handling COVID-19 that can empower the research community to develop new AI-based methods for the prediction of cases, diagnosis of patients, drug discovery, and design of vaccines.

Similarly to the survey of Lalmuanawma, Hussain & Chhakchhuak (2020) and Pham et al. (2020) we used the selective assessment method, while similarly to the survey of Ozsahin et al. (2020) we focused on less formal databases such as BiorXiv and ArXiv, which allows to analyzes the most recent trends in research without waiting for formal publication in indexation by major databases, which can take a significant amount of time. Due to the nature of this study being a relatively new research subject, we mostly focused on the pre-print papers.

The rationale for this survey is the extreme growth of research papers published on COVID-19 papers, which requires them to be analyzes, categorized, and reflected upon. Figure 1 shows the number of research papers posted on BiorXiv, ArXiv, and MedRxiv from January, 2020 to April 2021. This study is intended for researchers working in the intersection of computer science, artificial intelligence, and biomedical domains.

Figure 1 Number of preprints available related to COVID-19 on different platforms (From Jan, 2020 to April, 2021).

This survey covers literature published in 2020 (January–September) in different fields of study related to COVID-19. Because no similar efforts have yet been made, we draw on examples from the previous literature. Unbiased and comprehensive coverage of the previous literature was accomplished by querying online preprint research archives (BiorXiv, ArXiv, and MedrXiv) with search terms directly relates to COVID-19 (e.g., “covid-19”, “coronavirus”, “SARS-CoV-2”) and with Boolean operators to find studies applying artificial intelligence (e.g., “artificial intelligence” AND “neural networks”). Studies were included if they provided an overview of open science concepts that were relevant to addressing COVID-19. Once the papers were selected, they were categorized and analyzed based on their research aims and approaches used. Table S4 shows the selection and exclusion criteria for shortlisted research articles.

Coronavirus overview

A coronavirus is a group of viruses that can be transferred between human beings and animals. The novel coronavirus is known as SARS-CoV-2. Coronavirus (CoV) belongs to the family Coronavirinae of the order Nidovirales (Xie et al., 2020). CoV is broadly classified into four main classes namely α, β, γ, and δ. The first two (i.e., α and β) contaminate mammals only. The later ones (i.e., γ and δ) contaminate birds. They may contaminate mammals in a rare case. The genome of CoV consists of a single-stranded positive-sense RNA whose length is 30 kb (Chen, Liu & Guo, 2020). CoV is the largest among the existing RNA viruses. It has also a 5’ cap and 3’ poly-A tail (Chen, Liu & Guo, 2020).

Based on the literature available, CoVs infecting human beings include two α-CoVs (229E and NL63), and five β-CoVs (OC43, HKU1, MERS-CoV, SARS-CoV, and SARS-CoV-2). SARS-CoV-2 is a novel class of β-coronavirus genera that consists of bat-SARS-like (SL)-CoV ZC45, bat-SL-CoV ZXC21, SARS-CoV, and MERS-CoV. From recent studies, it found that SARS-CoV-2 came from wild animals. However, the exact source of this virus is unknown. There is a substantial genetic difference between now and SARS-CoV and numerous similarities among them in terms of chemical and physical characteristics (Su et al., 2016).

Data Gathering Systems for COVID-19

As we knew that the COVID-19 is an infectious disease. The spread of this virus can be stopped through no human interaction. AI-based tools are used to collect the COVID-19 data, develop a vigilant system, and visualize the COVID-19 data (Abd-Alrazaq et al., 2020). Recently, smartphone applications are also developed to diagnose the user’s health and trace the spread of infection. The main intention behind these applications is to identify vulnerable communities, provide real-time information to both patients and medical staff, detect infected hotspot areas, and generate advice for patient’s health (Naude, 2020). Figure 2 shows the categorization of data gathering systems for COVID-19.

Figure 2 Classification of data gathering systems.

An early cautionary and vigilant system

BlueDot is an effective analysis tool, which is developed in Canada (https://bluedot.global). It utilizes natural language processing (NLP) and machine learning technique. It was able to identify the outbreak of COVID-19 and generated vigilant alerts to users. It was used to generate the cautionary warnings to cities where the people reached from Wuhan city after January 2020 (Bogoch et al., 2020). In Belgium, the telecom operators are integrated with healthcare for analysing the infection spread in particular areas and identifying the infected hotspot areas. They have segregated the population into different regions according to the spread of infection. The same concept has also been used in some other countries. It provides real-time monitoring of patients and the consultant can use this information to prepare the prevention plans for virus infection in time (Yankoski, Weninger & Scheirer, 2020).

Similarly, Austria Telecom has agreed with its authorities to deliver the customer data. The customer data is used to trace their movements in the hotspot Lombardy region. MIT’s consortium is working on a smartphone application to detect the spread of virus infection. Global Positioning System (GPS) in a smartphone is used for checking the intersection of user’s trails with trails of infected persons. They used cryptographic techniques to protect the data. The application generates early cautionary signs to analyse the risk of infection spread after contacting with the infected persons. Instagram developed a COVID-19 tracker named as ‘RT.live’. It shows the state-by-state information on coronavirus infection in the US. The data analysis algorithm is used to estimate the reproduction of virus infection.

‘HealthyTogether’ application is developed by the Utah government for reducing the spreading of COVID-19. It is used to analyze the symptoms, determine the nearby testing center, and assessment of test results. Singapore Government agency developed a ‘TraceTogether’ application to protect the community from COVID-19. It helps contact tracers notify quickly through Bluetooth connection.

India Government developed the ‘Aarogya Setu’ application to monitor the infection caused by COVID-19 patients. This application has used a questionnaire to determine whether the person is infected or not. It is helpful to find if the infected patient was somewhere near the user. Another vigilant system was HealthMap to impose quarantines and restrict the movement of peoples.

Table S5 shows a brief description of the vigilant system for COVID-19. Table S6 depicts the functionality analysis of different real-time surveillance mobile apps for COVID-19 (Ming et al., 2020).

Data visualization systems

Data visualization techniques/dashboards utilized AI for tracking and forecasting of COVID infection. They provide a global overview of infected persons. Data visualization system is categorized into two broad categories namely local and global dashboards (Comba, 2020). The Center for System Science and Engineering of Johns Hopkins University developed the COVID-19 dashboard (JHU-CSSE) for global viewing of COVID-19. NextStrain generates the genomic epidemiology of COVID-19. It consists of a dropdown list for Asia, Africa, Europe, and Oceania. BBC dashboard provides the visualization of coronavirus infected areas in the World. It provides the effect of lockdown on the death cases in six countries namely the UK, Italy, Spain, France, Germany, and the US. The New York Times offers the dashboard for virus infection in various areas of the World (Anastassopoulou et al., 2020). HealthMap uses the World map for showing the visualization of infected cases in a particular area. Bing’s AI tracker is used to show infected, recovered, and fatal cases of different parts of the World. The filter is applied to a particular country to show the infected persons in a particular province. South Africa developed COVID-19 ZA South Africa dashboard for tracking the infected, recovered, and death cases in South Africa. It also tracks the number of tests conducted and positive cases in different provinces. Tableau implemented a COVID-19 Data Hub for a daily global tracker. COVID19 India is initiated by crowdsourcing. The infected, recovered, and deaths are shown in both tabular and graphical manner. Table S7 depicts the comparative analysis of different dashboards concerning the statistical information reported.

COVID-19 datasets and resources

Artificial Intelligence techniques require big data for further assessing the drug discovery, risk assessment, spread of infection, treatment, and cure of COVID infected patients. The datasets are broadly classified into text, social media, biomedical, speech, and case studies. The classification of COVID-19 datasets is shown in Fig. 3.

Figure 3 Classification of COVID-19 datasets.

Text datasets

The text datasets include risk factors, non-pharmaceutical interventions, incubation period, the spread of virus infection, and stability of the environment (Ahamed & Samad, 2020; Abd-Alrazaq et al., 2021). WHO Global Research Database provides scientific information on COVID-19. COVID-19 Open Research Dataset is an open dataset. The Kaggle challenges provide the COVID-19 data for analysis. LitCOVID and AI COVID-19 dataset consist of clinical trial data.

Social media datasets

COVID-19 Tweets and Covid-19 Twitter dataset comprise of coronavirus related tweets, which are misclassified information and rumors on Twitter (Lwin et al., 2020). These datasets contain the reactions from different persons on the tweets (Lamsal, 2020). COVID-19 Real World Worry Dataset (Kleinberg, van der Vegt & Mozes, 2020) consists of labeled texts of persons’ emotional responses towards COVID-19. It also consists of public sentiment and the mental health of persons in the infected areas of the World. Institutional, News, and Media Tweet Dataset consists of institutional communication of COVID infected areas in different countries. Both COVID-19 Coronavirus News Article dataset and COVID-19 Television Coverage dataset comprises of covering the news and prints of COVID-19 outbreak. These datasets cover the spreading of the virus, misinformation about drugs, and tracking of coronavirus.

Biomedical datasets

Covid Chest X-ray dataset is available as an open-source dataset. It consists of medical images that can be used to train deep learning/machine, learning models (Minaee et al., 2020). Data labeling and annotation can be used by radiologists. COVID-19 Survival Calculator uses the symptoms of infected patients. However, these are unmanageable and require manual maintenance. There are several data repositories, which provide links to researchers to easily access the data. Data4COVID Living Data and COVID-19 Dataset Clearinghouse repositories have many links to coronavirus data from different parts of the world. AI-based tools and techniques are used to extract the data from repositories, cleaning the data, and develop some predictions on cleaned data. The gene structure of SARS-CoV-2 should be known for the development of drugs and antibodies. Nextstrain is working on the gene structure of SARS-CoV-2 to illustrate the spread of coronavirus. RCSB Protein Data Bank and Global Health Drug Discovery Institute developed online COVID data portals for analysis of protein structure. American Chemical Society issued an open Covid-19 antiviral candidate compounds dataset that may be useful for the treatment of coronavirus infected patients. The crowdsourced developed a Fold.it (Miller et al., 2020) science game to design antiviral protein for fighting against COVID-19.

Speech datasets

Speech dataset consists of either cough sounds or breathing rate, which are used to detect the COVID-19. The smartphone and telemedicine can be used to collect the samples of cough and breathe of patients. Some researchers designed a cross platform for collection of both cough and breath. COVID-19 can be differentiated through the voice sounds. The scholars from the University of Cambridge made calls for voice sample collection from the participants. They made open source repository for cough samples.

Case studies

WHO and the National Centers for Disease Control released the COVID datasets for tracking the progress of infection and perceiving the effect of preventive measures. Johns Hopkins and Github (Xu et al., 2020) offer information about COVID-19 infected areas. Humanitarian Data Exchange hosted data about the coronavirus. CHIME is a COVID-19 Hospital Impact Model for Epidemics based SIR modeling that utilizes the number of infected, recovered, and fatality persons to predict the infection caused by coronavirus and the requirement of hospital beds. Table S8 depicts the downloadable links and description of datasets (Shuja et al., 2021).

Prediction Models

AI-based prediction models can be used to predict the mortality rate, infection in patients through cough and biomarkers, genome structure associated with fatality rate, and severity rate. The epidemic of the tracker was developed by Metabiota. It utilized the forecasting machine learning model used for the prediction of infection spread. Robert Koch Institute developed a SIR prediction model, which uses quarantines, social distancing, and lockdowns. This model was implemented in the R language. It was helpful to reduce the spreading of infections.

Yan et al. (2020) proposed a machine learning technique to predict the severity of COVID-19. The survival rate of patients at Tongji Hospital in Wuhan is assessed through the prognostic biomarker. The prediction accuracy obtained from the proposed machine learning technique was approximately 90%. Jiang et al. (2020) developed an AI framework that has the predictive capability to analyze the severity of patients. They developed an algorithm to identify the clinical characteristics of infected persons. The predictive model used clinical characteristics to predict the severe illness of 53 patients with 80% accuracy. The predictive model is tested on two hospitals in Wenzhou, Zhejiang, China.

Alotaibi, Shiblee & Alshahrani (2021) used machine learning techniques namely Support Vector Machine (SVM), Artificial Neural Network (ANN) and Random forest to predict the severity of infected patients. The prediction accuracies obtained from SVM, ANN, and Random forest are 86.67%, 83.33% and 90.83%, respectively. Chatterjee et al. (2020) developed a susceptible, exposure, infectious and recovered (SEIR) model to study the impact of healthcare during the pandemic situation. By using this model, hospitalizations and ICU requirements can be reduced to 90%. Ghosal et al. (2020) used a linear regression to predict the number of deaths in India. The predicted death rate is 211 and 467 by the end of 5th and 6th week, respectively.

Imran et al. (2020) developed the AI4COVID-19 tool for the preliminary diagnosis of COVID-19. AI4COVID used a 2-second cough recording of infected persons. After the analysis of cough samples, the AI tool generates preliminary diagnosis for patients. AI4COVID distinguished COVID and non-COVID patients with 90% accuracy. Feng et al. (2021) developed a diagnosis model for the preliminary identification of infected patients. The prediction was based on travel history, clinical symptoms, and test results. Lasso regression applied to features obtained from clinical symptoms for prediction. The features are extracted from Rao & Vazquez (2020) implemented machine learning algorithms to determine the possible causes of coronavirus in the quarantine areas through a mobile-based survey. The algorithms can easily predict the no-risk, moderate risk, and high risk of infection through travel history and contact with infected persons.

Tang et al. (2021) implemented a machine learning technique for automatic severity assessment of infected patients using chest CT images. They extracted features from lungs and applied them to a random forest model for predicting the severity of COVID-19. The accuracy obtained from the random forest model was 87.50%. Yue et al. (2020) used logistic regression (LR) and random forest (RF) to extract the features from 72 pneumonia lesions of 31patients. The sensitivity and specificity obtained from LR are 1.0 and 0.89, respectively. However, the RF model provided a sensitivity of 0.75 and a specificity of 1.0. Patrikar et al. (2020) modified the SEIR framework to study the effect of social distancing on the spread of coronavirus. They found that the infection can be reduced to 78% through social distancing.

Table S9 shows the comparative analysis of prediction models for COVID-19.

Screening Techniques for COVID-19

The diagnosis of a COVID-19 patient is a challenging task. The testing of every patient is time-consuming. Due to the pandemic situation, faster and cheaper tests are required to generate the medical report. AI-based techniques are used to screen COVID-19 patients. There are three different methods to screen patients using AI. These are face recognition, wearable devices, and virtual healthcare assistant (see Fig. 4) (Ting et al., 2020).

Figure 4 Screening techniques for COVID-19 symptoms.

Screening through patients’ face

Face recognition technology is used to eliminate the spread of infection caused by coronavirus through human contact. It can be used with a temperature detection tool for the efficient identification of COVID-19 patients. Drones/Robots equipped with the thermal scanner are used for detecting fever in patients from an appropriate distance. Chinese firm Baidu developed infrared cameras to scan the crowds for temperature scanning (Martin & Lin, 2020). They can screen hundreds of persons within one minute. However, the fever can be wrongly detected if a person wears some items on the face. AI-based face recognition tools can be installed at schools, colleges, railway, airport, and community places. These tools can automatically detect the persons having fever, tracing their positions, and detect whether the person has a mask or not.

Screening through a wearable device

Nowadays, wearable health devices such as Fitbits and Garmins are more popular to monitor physiological parameters such as heart rate, blood pressure, oxygen levels, body temperature, movement, and sleep for a better lifestyle. Apple developed an AI-based watch to determine the temperature and heart rate to identify the symptoms of patients (Xavier, 2020). OURA developed an activity tracking ring that uses body temperature, breathing rate, and heart rate to determine the onset patterns, progress, and recovery of the patient (Robitzski, 2020). Stanford Medicine and Google company came together to utilize the data collected from the wearable device to detect the symptoms of an infected person. They used body temperature and heart rate for fighting against the COVID-19 infection. Central Queensland University collaborated with Cleveland Clinic to analyze the data collected from Whoop’s wearable devices (Hirten et al., 2021). Shanghai Public Health Center has developed in-built temperature sensors in wearable devices to detect the body temperature for COVID-19 patients regularly. The detected temperature is continuously sent to the nursing station for patient monitoring. Canada based Proxxi technologies developed a wrist device named ‘Halo’ that communicate to user through vibration (Russey, 2020). It gives an alert to the user about his come within range of 6 feet of another wearable user. It uses Bluetooth technology to communicate with others and keep the data record about users whom the wearable user met.

Screening through Chatbots

In this pandemic situation, chatbot developers and healthcare systems are integrated for the prediction of infected patients. Chatbots are used as an accelerator tool for the healthcare of COVID infected persons. AI-based Stallion used the capabilities of natural language processing (NLP) to develop a Chatbot as a virtual healthcare agent (Obeidat, 2020). It endorses protection measures, monitors symptoms, and generates suggestions to individuals for home quarantine or hospital admissions.

Some countries developed a “Self-Triage” system that uses a questionnaire about the symptoms of patients. Microsoft developed a healthcare virtual assistant that will help to determine the appropriate action using the symptoms of patients. They included risk assessment, clinical triage, and COVID-19 question-answering in their chatbot (Schreurs, 2020).

Google Cloud developed a virtual assistant that will provide information about COVID-19. The virtual assistant quickly responds to questions raised by users and provide optimal information to users (Maddox, 2020). BITS students developed a chatbot to yield awareness among the users. AI-enabled doctor video bot, named AskDoc, is developed to provide answers about the COVID-19 queries using voice and text. Facebook implemented a WhatsApp bot to support the users to update them on the outbreak of COVID-19 (Hutchinson, 2020). Zoe chatbot gives users answers and appropriate information (Chanthadavong, 2020).

Table S10 shows the AI-based tools/techniques for the screening of COVID-19 symptoms.

Diagnosis of COVID-19

In this pandemic situation, a quicker diagnosis is required. The most widely used technique for diagnosis is a real-time reverse transcription-polymerase chain reaction (RT-PCR) (Long et al., 2020). The well-known radiological imaging techniques are X-ray and computed (CT) (Zu et al., 2020). Due to less sensitivity (60%-70%) of RT-PCR, symptoms can be detected through radiological images (Kanne et al., 2020). However, radiological images are sensitive to detect the infection caused by COVID-19 and can be used to monitor the patients (Lee, Ng & Khong, 2020). A less number of clinical expertise is available as compared to the COVID-19 cases that arose in this pandemic situation (Wynants et al., 2020). Therefore, AI-based tools and techniques can be used for faster diagnosis.

Radiological imaging

Researchers studied 33% of chest CTs have rounded lung opacities. It is observed from the CT scan that symptoms may not be detected in the initial 2 days (Bernheim et al., 2020). The abnormal finding is detected in the CT scan of patients after 10 days of symptoms observed (Pan et al., 2020). Initially, due to the low sensitivity of RT-PCR kits, clinical experts suggested using Chest CT for diagnosis (Ai et al., 2020). The traditional methods take 15 minutes for analysis of the chest CT scan. Nowadays, machine learning or deep learning techniques are used for automated analysis of CT scans and chest X-rays (Lopez-Cabrera et al., 2021). These techniques will help to speed up the analysis process (Ng et al., 2020; Fang et al., 2020; Weinstock & RJea Echenique, 2020). Figure 5 depicts the distribution of radiological images for diagnosis of COVID-19.

Figure 5 Different radiological modality used in deep learning techniques.

AI tools for radiological images

Baidu’s team developed a LinerFold software that diagnoses the infection of COVID-19 in 27 s (McFarland, 2020). The prediction time is reduced from 55 min to 27 s and helps in developing the drug for coronavirus. It can identify lesions in terms of volume, proportion, and numbers. The accuracy obtained from the system is 92% on available datasets. China scientists developed a healthcare application named InferVISION to investigate COVID-19 patients. InferVISION utilized NVIDIA’s Clara SDK (Chen, 2020). It can identify the positive cases within a very small amount of time. Shenzhen-based company has developed a MicroMultiCopter that can carry medical samples from infected and dense areas. It can also be used for food and medical items delivery. LinkingMed technique to analyze the CT scan in less than 60 s. It provides 92% accuracy on test datasets.

Canadian based DarwinAI developed a neural network to analyze the X-rays for COVID-19 infection. Some hospitals do not have testing kits and radiologists to analyze the infection. In that case, an X-ray is an alternative to testing kits. DarwinAI developed a COVID-Net at University of Waterloo. DarwinAI trained from 17,000 X-rays images. They are working on COVID-Net for identifying the risk of infection associated with workers (Borkowski et al., 2020).

Mumbai-based company Qure.ai developed an AI-based chest X-ray system named qXR (Bora, 2020). The qXR is used to detect COVID patients from chest X-rays. qXR utilized deep learning models to detect lung abnormalities. It will help trainee doctors for their second opinion about the patient. The accuracy obtained to detect the COVID infected patients is approximately 95% over 11,000 patients. Lunit’s software named “INSIGHT CXR” is used to scan the abnormalities of the lungs (Lunit, www.lunit.io/covid19). These tools will help to handle the coronavirus pandemic.

Ron Li implemented an Epic model that can assess whether the patients have to shift to ICU or not (Strickland, 2020). He explored the “Deterioration Index” to identify whether the patient’s condition is deteriorating or not. Epic trained 130,000 patients for assessing the validity of the Deterioration Index. He modified the model for the evaluation of COVID-19 patients in March. Six different organizations have been evaluated the performance of the Epic model on 3,000 COVID patients and proved its performance.

Johns Hopkins University (JHU) developed a diagnostics tool for coronavirus infection (Strickland, 2020). Researchers in JHU developed a vigilant system for respiratory failure that can be caused by COVID. The respiratory diagnosis model will help the doctors to assess the infected patients. It will also envisage the need for ventilators and critical hospital instruments.

Maghdid et al. (2020) proposed a mobile application to scan the CT images. CAD4COVID is an AI-based software to distinguish infected patients from chest X-rays (Delft Imaging, 2020). Huazhong University of Science and Technology developed an algorithm for estimation of COVID-19 infected person with 80% accuracy. However, they tested on 53 patients of two different Chinese hospitals. Table S11 depicts the description of AI tools used for the treatment of COVID-19.

Deep learning architectures for radiological images

Hemdan, Shouman & Karar (2020) developed an automated diagnose framework called COVIDX-Net for the analysis of X-ray images. COVIDX-Net utilizes seven different deep learning models and tested over 50 X-ray images. The accuracy obtained from COVIDX-Net is 90%. Wang, Lin & Wong (2020) developed a deep convolutional neural network (CNN) model (COVID-Net) for the identification of infection in chest X-ray images. COVID-Net model tested over 13,800 chest X-ray images and obtained 93.3% accuracy in recognizing normal, typical pneumonia, and COVID-19 cases. Apostolopoulos & Bessiana (2020) evaluated CNN for the classification of COVID-19 cases. They used the transfer learning technique on 1427 X-ray images, achieving 98.75% and 93.48% accuracy for two and three class classification, respectively. Khan, Shah & Bhat (2020) proposed a CoroNet architecture for detection of COVID-19 in chest X-ray of infected patients. CoroNet utilized the concept of Xception model. The classification performance of CoroNet was 99% and 89.6% for binary and three-class, respectively.

Ozturk et al. (2020) developed a DarkCovidNet model for the detection of infected patients using chest X-ray images. DarkCovidNet model has seventeen convolutional layers and different filtering for each layer. The classification accuracies obtained from DarkCovidNet were 98.08% and 87.02% for binary and multi-class, respectively. Sethy et al. (2020) suggested deep learning based methodology for detection of infected patients using chest X-ray images. The proposed methodology used CNN models with support vector machine (SVM). The classification accuracy of ResNet50 model with SVM classifier is 95.38%. To overcome the shortcoming of hyper-parameter tuning associated with transfer models, Kaur et al. (2021) utilized the strength Pareto evolutionary algorithm-II (SPEA-II) for chest X-ray images. They modified AlexNet to extract the features from X-ray images and applied on classification process. The proposed approach outperforms the competitive models in terms of performance measures. Jain et al. (2021) developed a deep learning model for diagnosis of chest X-ray images. IncpetionV3, Xception, and ResNeXt are used in the development of proposed model. The classification accuracies obtained from this model are 99% and 96% for training and testing, respectively.

Chowdhury et al. (2020) designed an automatic coronavirus detection technique from X-ray images. This technique was evaluated on Chest X-ray dataset and attained the classification accuracy of 99.7%. Islam, Islam & Asraf (2020) combined convolutional neural network (CNN) and long short-term memory (LSTM) to diagnose coronavirus infection in the patients. The performance of this model was validated on 4575 X-ray images. The sensitivity and specificity of this model were 99.3% and 99.2%, respectively. Nour, Cömert & Polat (2020) developed a CNN model to extract discriminative features from chest X-ray. These features were applied on three well-known machine learning algorithms namely, k-nearest neighbor (KNN), SVM, and decision tree for classification. The sensitivity and specificity obtained from SVM-based classifier are 89.39% and 99.75%, respectively. A number of research articles were published in the field of diagnosis of COVID-19 using chest X-ray images (Oh, Park & Ye, 2020; Pereira et al., 2020; Elaziz et al., 2020; Jain et al., 2020; Panwar et al., 2020; Tsiknakis et al., 2020; Gianchandani et al., 2020; Singh et al., 2021; Das et al., 2020; Altan & Karasu, 2020; Ghaderzadeh & Asadi, 2021).

Wang et al. (2021) implemented deep learning approaches to extract specific features from CT scan images. These features are used to detect coronavirus infection using the transfer learning model. The developed model is tested on 1,065 images of COVID-19. The accuracy obtained from their model is 89.5%. Tan et al. (2020) hybridized super resolution generative adversarial network (SRGAN) model and VGG16 to detect infected patients by chest CT. SRGAN was used to enhance the resolution of CT images. VGG16 was used to differentiate the infected and healthy region of CT. The developed model is validated over 275 COVID-19 and 195 normal CT images. The classification accuracy obtained from the model was 97.87%. Xu et al. (2020) used different CNN models to detect infected patients. CT images are processed to extract the interesting regions. 3D CNN model is used to segment the CT images. Thereafter, the segmented images are further classified into three different classes. The overall accuracy of deep learning models was 86.70%.

Singh et al. (2020) developed an automatic chest CT analysis system to classify the infected persons whether these are positive or not. The CNN hyper-parameters are tuned through multi-objective differential evolution (MODE). CNN with optimized parameters is used for COVID-19 patient classification. The proposed model outperforms the other competitive models by 1.927%. Li et al. (2020) developed a COVNet model for extracting 2D and 3D global features from chest CT for the classification of COVID-19 patients. The extracted features are utilized to differentiate CoVID-19 infected, non-pneumonia, and community-acquired pneumonia (CAP). The classification accuracy obtained from COVNet was 96.00%. Mei et al. (2020) utilized machine learning techniques for the analysis of chest CT scans of COVID-19 patients. The clinical symptoms and laboratory testing were integrated with CT scans for analysis.

Hasan et al. (2020) combined both deep learning technique and Q-deformed entropy for classification of COVID-19 using CT scan images. The features were extracted from CT scan images using Q-deformed entropy and CNN. The extracted features were applied on LSTM for distinguishing the COVID-19 and non COVID-19. The proposed approach attained the classification accuracy of 99.68%. Wu et al. (2020) developed a multi-view fusion model for identification of coronavirus infection in the CT scan images. This model was evaluated on the CT images of 495 patients. This model takes approximately ten minutes for analysis of CT images of infected patients. They used Youden index for fusion process. For testing phase, the sensitivity and specificity obtained from multi view model are 81.1% and 61.5%, respectively.

Ko et al. (2020) developed a FCONet model for COVID-19 classification using chest CT. FCONet utilized VGG16, ResNet50, InceptionV3, and Xception. The performance of FCONet was validated on 3,993 chest CT images. The accuracy obtained from ResNet50 was 96.97%. Researchers use deep learning architectures for analysis of CT scan images (Jaiswal et al., 2020; Singh, Kumar & Kaur, 2021).

The comparative analysis of deep learning techniques on radiological images such as chest X-ray and CT scan images are illustrated in Tables S12 and S13, respectively, while the usage of deep learning models is summarized in Fig. 6.

Figure 6 Different deep learning architectures used for diagnosis of COVID-19.

Non-invasive techniques

Several techniques do not need any specialized radiological equipment for diagnosis and treatment of COVID-19. Cho et al. (2014) used a GRU neural network to determine the respiratory patterns of patients. The developed model is trained on footage obtained from Kinect depth cameras (Wang et al., 2020) and recognizes the COVID-19 patients.

Cascella et al. (2020) suggested that COVID-19 patients have respiratory patterns, which are different from the common cold and flu. However, the abnormal respiratory patterns have no direct correlation with the diagnosis and treatment of COVID-19. The wearable devices and mobile applications can be utilized in the diagnosis and treatment of COVID-19. These devices and apps may utilize the body temperature, heart rate, cough samples, and breath rate.

Drug Design

AI has the potential to discover, design, and repurpose the existing drugs to combat the COVID-19 as shown in Fig. 7. During this pandemic situation, several research laboratories are trying to develop vaccines/drugs against COVID-19. They are using AI techniques to discover new vaccines or repurposing existing drugs (Kaushik & Raj, 2020).

Figure 7 Deep learning and machine learning based drug synergy prediction for COVID-19.

Envisaging virus-host interactome

The prediction of protein structure is necessary to develop new drugs. Senior et al. (2020) developed an AlphaFold model to envisage the structures of proteins associated with SARS-CoV-2. ResNet is used to extract the features from amino acid sequences (Jumper et al., 2020). Heo & Feig (2020) implemented dilated ResNet to envisage the protein structure of SARS-CoV-2. They refined the AlphaFold’s predicted structures using molecular dynamics. Ge et al. (2020) suggested an approach to construct a knowledge graph that involving human proteins, viral proteins, and drugs. The knowledge graph is used to envisage possibly candidate drugs. Nguyen et al. (2020) developed a SARS-based model using mathematical deep learning to determine possible inhibitors. 84 SARS coronavirus inhibitors are envisaged from ChEML and PDBind databases. Zhou et al. (2020) developed network-based model to repurpose the drugs for SARS-CoV-2. Hu, Jiang & Yin (2020) implemented a neural network to predict the affinities of SARS-CoV-2 proteins. They found ten possible drugs with their binding affinity scores among 4,895 drugs.

Exscienta designed an AI-based drug molecule for coronavirus as reported in news (Kirk, 2020). It is the first company who has designed the drug molecule for coronavirus. The traditional drug discovery research took 4–5 years for developing new drugs. However, it will take 1 year to develop the molecular structure. Insilco Medicine company used generative adversarial networks to determine the molecule structure. This structure is used to discover drugs. Insilco company screened 100 molecules for synthesis and testing. AI can be used to develop antibodies and vaccines for COVID-19 (Vanderslott, Pollard & Thomas, 2020). It can be done in two ways namely, from scratch and drug repurposing. Google’ DeepMind developed an AlphaGo algorithm to envisage the protein structure of the virus, which can help develop new vaccines against coronavirus (Silver et al., 2018). However, the experimentation has to be performed for validation of the designed protein.

Envisaging interaction among coronavirus and drugs

AI can be used to screen the exiting drug molecules and find their suitability against the coronavirus. South Korea and the USA used an AI-based algorithm to identify the ‘Atazanavir’ drug for repurposed to the treatment of COVID-19 (Beck et al., 2020). Researchers of Benevolent AI identified ‘Baricitinib’ and ‘Myelofibrosis’ drugs for the treatment of COVID-19 (Stebbing et al., 2020). Singaporean Firm Gero used a deep learning technique to recognize ‘Afatinib’ for the treatment of COVID-19. Zhang et al. (2020) used fully connected ANN to envisage binding affinities from the PDBbind database. They explored the existing molecules for the treatment of SARS-CoV-2 (Zhang et al., 2020). Beck et al. (2020) developed Molecule Transformer-Drug Target Interaction (MT-DTI) model to determine antivirals drugs that may be effective against coronavirus. BERT algorithm used to compute the binding affinities of existing drugs. Hofmarcher et al. (2020) used Long Short-Term Memory (LSTM) model on SMILES data to screen 900 mln compounds from the ZINC dataset. 30,000 possible compounds are selected for treatment of SARS-CoV-2. These treatments may be available in near future. The main reason behind this that medical trials, checks, and control are needed before the approval of these drugs. After the identification and screening of drugs, a vaccine formulation may take a minimum of 18 months (Levin et al., 2020).

Social Implications

Nowadays, the different myths regarding the incorrect data of infection, unsuitable drugs, and misclassified infected zones have been propagated on social media. However, these rumors and hate speech can be greatly affected by the social life of human beings. WHO put these things in infodemic, i.e., the huge amount of data is available with a mixture of accurate and inaccurate data. Due to this, persons are unable to find verified sources whether the given information is correct or not in this pandemic situation.

Analysis of social media

In this epidemic situation, social media should assess the quality and accuracy of the information posted. Nowadays, Facebook and Google are working against the misinformation, phishing funding websites, and viruses floating on their platforms. When you search the COVID on YouTube, then it links to the user either government organization or WHO for retrieving correct information. Videos posted on YouTube are screened and dropped immediately from the site after the false information is confirmed. Eichstaedt et al. (2015) analyzed the tweet posted on Twitter by the user during this epidemic situation of COVID-19. AI-based text analyzer utilizing the number of cases and death in a particular region to investigate the mental health of the user. MIT developed a neural network-based model to determine the effectiveness of quarantine measures and the spread of virus infection (Dandekar, Rackauckas & Barbastathis, 2020). Khataee et al. (2020) reported the effects of social distancing during the COVID epidemic. They can assess the local elements of the pandemic. They reported that the US has not taken precautionary measures to stop infection caused by a coronavirus. Rosenberg, Syed & Rezaie (2020) analyzed the tweets related to coronavirus-related information. They explored the various myths about the virus. Gallotti et al. (2020) explored the social media tweets posted on Twitter. They developed an Inodemic Risk Index (IRI) to determine verified human beings, unverified human beings, verified, and unverified bots. IRI utilizes the number of users and messages posted by users and their reliability. They highlighted the impact of infodemics, social outcomes, and controlled pandemics. Cinelli et al. (2020) studied the contents of social media. They assessed the development of the discourse on Twitter, YouTube, Instagram, and other social media. They analyzed the comments, likes, and action upon comments for 45 days. Mejova & Kalimeri (2020) studied the advertisements related to coronavirus posted on Facebook. Facebook Ad Library is used to examine all the advertisements that have the phrases “coronavirus” and “covid” across the world. They established that 5% of advertisements having misclassified and misconception information. Zarocostas (2020) studied the information shared and posted on different social media regarding COVID-19. AI tools can be used to track the spreading of rumors regarding the coronavirus. Pandey et al. (2020) reported the breach in delivering genuine information to users in India. They retrieved the information and found their verified sources using artificial intelligence and NLP. They tested their approach on Sanitation and Hygienic information for this epidemic situation. AI-based chatbots can be used to propagate COVID-19 information that can filter the misinformation. WHO developed a multilingual chatbot to explore the information posted on social media and news channels (Miner, Laranjo & Kocaballi, 2020). This virtual assistant can be used to verifying the information before further processing.

Analysis of hate speech

Hate speech is a major concern in the last few months. The verbal and non-verbal abuse statements may give rise to physical violence against the corona warriors. Such instances were reported in the lockdown situation of various countries. Velásquez et al. (2020) analyzed the hate speeches regarding COVID-19 posted on different social media and their movement from one media to another. They have also analyzed the methods for transmission and found that hate speeches are rapidly spread in the epidemic situation of COVID. Schild et al. (2020) reported the Sinophopic behavior of tweets posted on Twitter and other media. They trained machine learning models on the information obtained from the contents of COVID-19. Web can be used for spreading misinformation and hate speech on COVID-19 information. AI-based tools play a vital role in fighting against hate speech (Atehortua & Patino, 2021).

Table S14 depicts the impact of rumors and hate speech on social life.

Future Research Directions

The possible research directions for application of AI on COVID-19 are described below:

Interpretable prediction

The results obtained from the AI-based prediction model should be interpretable and easy to use (Roberts et al., 2021; Wieczorek, Siłka & Woźniak, 2020). Therefore, soon one can utilize information extraction or image captioning kin of techniques to provide more interpretable prediction results.

Mobile-based AI tools

The AI based models can be deployed on lightweight devices such as mobile. Our objective is not to achieve diagnostic tools for hospitals and clinics. On can work on AI-based COVID-19 prediction and diagnostic tools on lightweight devices (Iyengar et al., 2020).

Drug discovery

Till now many researchers have worked on developing drugs for COVID-19 infected patients. However, no effective drug is available for COVID-19 diagnosis. Therefore, one may utilize the existing medicines to build an efficient drug for COVID-19 infected patients (Ho, 2020).

AI-based drones to combat COVID-19

As we knew that the main source of infection transmission is human contact. Due to this, AI-based robots are used to disinfect the patient’s rooms and interact with patients. For disinfection, robots are transmitting ultraviolet light over infected space to remove the virus. Robots are transferred the face image and voice of doctor on their screen during the interaction with patients. Hence, the medical staff is safe due to less contact with patients. Drones are widely used for transferring patient samples and medical apparatus. These are also used to disinfect the unreachable infected areas. During the lockdown, the drones are used to track the person who is come out from home. It provides faster delivery and less risk of infection. The drones supported by AI and computer vision techniques can play an essential role in the aerial monitoring of disease spread; for logistics and medical supply delivery (Angurala et al., 2020), as well as for social distance checking (Kumar et al., 2021). They also were used to perform aerial spray and disinfection of residential areas (Yaacoub et al., 2020). However, there are data security, and privacy concerns that need to be resolved for successful application of AI-supported drone technology in public spaces on a large scale.

3D Printing techniques for developing COVID-19 prevention and fighting tools

The 3D printing techniques may be used to design the face masks and face shields to be used for protection against COVID-19 (Swennen, Pottel & Haers, 2020). Combined with face-scanning technology and computer-aided design (CAD) tools these can be made to fit the individual face and head measurements. The 3D techniques also can be used to build other COVID-19 fighting tools such as main components of respiratory support equipment (Tino et al., 2020). These components can be designed using low-cost consumer filament extrusion printers. The 3D printing technology can quickly address the deficiencies of medical materials and spare parts of medical equipment, however, the processing time, high cost, and lack of manpower can be potential barriers for applying 3D printing on a larger scale (Ishack & Lipner, 2020). AI techniques can play a role in optimizing the 3D design process and reducing the cost of printing (Wang et al., 2020; Longhitano, Nunes & Candido, 2021; Almalki & Azeez, 2020).

Findings, Lessons and Recommendations

However, numerous challenges and research limitations have been indicated in the academic literature and need to be addressed in the future. Some of these challenges are related to nature and behaviour of COVID-19 because understanding how the virus spreads and how people can be infected caused by the complexity of this epidemic disease is extremely difficult. The lack of large-scale datasets in the academic literature for COVID-19 is considered a challenging task for AI researchers because it hinders the understanding of viral patterns and features. In order to make AI and big data platforms and applications a trustful solution to fight the COVID-19 virus, a critical challenge is the collection of large-scale datasets and making them open for research.

AI and big data-based algorithms should be optimized further to enhance the accuracy and reliability of the data analytics for better COVID-19 diagnosis and treatment. AI is able to provide viable solutions for fighting the COVID-19 pandemic in several ways. For example, AI has proved very useful for supporting outbreak prediction, coronavirus detection as well as infodemiology and infoveillance by leveraging learning-based techniques such as ML and DL from COVID-19-centric modeling, classification, and estimation. Moreover, AI has emerged as an attractive tool for facilitating vaccine and drug manufacturing. By using the datasets provided by healthcare organizations, governments, clinical labs and patients, AI leverages intelligent analytic tools for developing effective and safe vaccine/drug against COVID-19, which would be beneficial from both the economic and scientific perspectives. Moving forward, it is imperative for AI-designers and researchers to work together with medical professionals to create and develop these systems that are applicable to real-world datasets

Conclusions

In this paper, the impact of artificial intelligence (AI) techniques is assessed on early cautionary and vigilant systems that focus on COVID-19 warning and diagnostics. The COVID-19 datasets and visualization techniques are discussed with their applications. The AI-based diagnosis and treatment of COVID patients are assessed. The impact of AI on drug discovery and design is evaluated. The effect of COVID-19 is also evaluated on social and economic aspects. From the literature review, we can conclude that the AI techniques are widely used to identify the novel drug discovery beforehand of the outbreak of COVID-19. The AI techniques can help to search for the optimal drug against COVID-19. AI can be used to build the biomedical knowledge structure that connects the drugs and viruses to repurpose the existing drugs that are used to treat other diseases. The existing drug and coronavirus interaction can be modelled by AI techniques. Other promising applications of the AI methods are contact tracing, drone-based surveillance, and smart face mask design.

Supplemental Information

Supplemental Information 1 Symptoms associated with COVID-19 patients.

Click here for additional data file.

Supplemental Information 2 Estimation of severity of COVID-19 patients.

Click here for additional data file.

Supplemental Information 3 Summary of related survey papers.

Click here for additional data file.

Supplemental Information 4 Selection and elimination criterion for shortlisted research articles.

Click here for additional data file.

Supplemental Information 5 Vigilant systems for COVID-19.

Click here for additional data file.

Supplemental Information 6 Qualitative analysis of different surveillance mobile apps for COVID-19.

Click here for additional data file.

Supplemental Information 7 Comparison of Data Visualization Systems.

Click here for additional data file.

Supplemental Information 8 Description of COVID-19 datasets and resources.

Click here for additional data file.

Supplemental Information 9 Comparative analysis of prediction models.

Click here for additional data file.

Supplemental Information 10 Artificial Intelligence Tools/Techniques for screening of COVID-19 symptoms.

Click here for additional data file.

Supplemental Information 11 AI tools for diagnosis of COVID-19.

Click here for additional data file.

Supplemental Information 12 Deep learning/Machine learning techniques used for Chest X-ray images.

Click here for additional data file.

Supplemental Information 13 Deep learning/Machine learning techniques used for Chest CT-scan images.

Click here for additional data file.

Supplemental Information 14 Impact of rumors and loathe speech on Social Life.

Click here for additional data file.

Additional Information and Declarations

Competing Interests

Author Contributions

Data Availability

Robertas Damasevicius is an Academic Editor for PeerJ.

Vijay Kumar conceived and designed the experiments, performed the experiments, analyzed the data, prepared figures and/or tables, authored or reviewed drafts of the paper, and approved the final draft.

Dilbag Singh conceived and designed the experiments, performed the experiments, analyzed the data, prepared figures and/or tables, authored or reviewed drafts of the paper, and approved the final draft.

Manjit Kaur conceived and designed the experiments, performed the experiments, analyzed the data, prepared figures and/or tables, authored or reviewed drafts of the paper, and approved the final draft.

Robertas Damaševičius analyzed the data, prepared figures and/or tables, authored or reviewed drafts of the paper, and approved the final draft.

The following information was supplied regarding data availability:

This is a survey article. No data or code was created during the writing of this article. The only resources used are the articles listed in the References section.

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
