# Peer review of "Overview of current state of research on the application of artificial intelligence techniques for COVID-19"

_PeerJ Computer Science, doi:10.7717/peerj-cs.564_

## Round 0.1 · original submission · Major Revisions

While reviewers value the timeliness of this survey paper, they also believe that it still needs substantial improvements before it can be considered for publication, particularly in the case of reviewer #3. Most importantly, I would highlight the following major issues, but please read and consider carefully the reviews provided by the authors:

- More detail is needed on the description of the methods and which ones lead to better performance.
- A more critical analysis of findings and results is missing.
- The paper needs to be proofread to correct typographical errors and improve grammar.
- The paper needs to be distinguished from other similar reviews.
- The paper should more carefully address the paper selection criteria and whether non-peer-reviewed papers have been considered.

Reviewer 1 ·

Basic reporting

The article presents an overview of recent papers which use artificial intelligence methods for prediction and diagnosis of COVID-19. Since there has been many such papers published in recent months, an overview is timely and reflects the need to summarize the existing research. The presented overview is broad and comprehensive, citing 142 literature sources. Such overview papers are highly needed by the researchers trying to orient themselves in the deluge of results reported on COVID-19. Clear English is used throughout. Professional article structure, figures and tables are provided.

Experimental design

The description of the methodology of survey should be improved to allow for replication. Specifically, indicate the query string. Were any papers found excluded from further analysis and why? Why you analyze only Preprints rather than complete publications registered on scientific bibliography databases such as Web of Science or Scopus?

Validity of the findings

The findings are valid, however, they must bus discussed in more-detail as suggested in my Comments for the author.

Additional comments

I suggest the following minor comments to be addressed before the paper could be accepted for publication:
1. Present a more extensive bibliographic analysis of literature sources on COVID-19, including the most popular venues of publications and leading research teams around the world. You can use, for exampling, the guidelines presented in Conducting systematic literature reviews and bibliometric analyses, Australian Journal of Management 45(2):175-194, DOI: 10.1177/0312896219877678 on how to visualize the results of literature reviews.
2. I suggest to present a summary for each type of system or application discussed (such as including information on the number of instances and attributes for biomedical datasets; or what are the most popular deep learning architectures used). This would provide a deeper insight on the analyzed research domain.
3. Add a critical discussion section and discuss the current limitations of such systems as well as the successes.
4. Present a more extensive and in-depth conclusions of where the COVID-19 related research is heading.

Reviewer 2 ·

Basic reporting

This paper surveys contributions of AI to COVID-19. It groups the surveyed articles into five categories: AI systems that predict the COVID-19 disease, diagnose the disease, recommend treatment, design new drugs, and predict social and economic impact. Therefore, the title of the paper is somewhat misleading as it spans more than prediction and diagnosis. It is therefore recommended to change the title.
The survey is definitely an interesting idea and could be of interest to many readers.
In the data gathering part, the grouping of different data gathering and prediction systems is interesting. It is recommended to refer to “surveillance” systems and to epidemiology for those forecasting the spread of the disease.
Regarding the form, there are many spelling and grammatical errors in the paper. It is recommended to have the paper proofread by a native speaker.

Experimental design

The methodology of querying only pre-print papers may not be sufficient to cope with the extent of the literature in this domain, so that it is recommended to broaden the base of papers studied. However, the information collected from the body of literature is already interesting so that the extension to a broader literature could wait until a next paper. This should be discussed in future directions.

Validity of the findings

One general impression reading about the different AI systems presented is that we do not have much data regarding their effectiveness. They definitely aim at addressing diverse challenges related to COVID-19, but do not have measurements showing that they do so. This is expected due to the novelty of the disease, but it should be better explained in the paper. You are proposing ideas for addressing the challenge, but these ideas need to be tested to be shown effective.
It is recommended to better highlight the systems that being used or are going to be used in medicine or in the field, by contrast to the systems only used by computer systems on existing datasets. The reader would like to know, among these systems, which ones actually advance the diagnosis, treatment, prediction etc. of COVID-19 on real persons, not only on existing datasets.

Additional comments

The paper is interesting to read and covers a broad range of AI applications in the fight against COVID-19. The language should be improved before publication, and the recommendations listed above should be considered.

Reviewer 3 ·

Basic reporting

• The review is riddled with typographical and grammatical errors that are too numerous to detail here. Furthermore, the writing is not professional enough to meet standards for an international audience. While I am sympathetic towards the barriers that non-native English speakers have to overcome, these language issues actually hampered my ability to follow the paper. I strongly advise having a native English speaker proofread the review.
• Introduction and background are adequate but suffer from logical gaps that weaken the section. For instance, in line 58, outcome risk is not clearly defined. Instead a confusing sentence is presented, “If the treatment is not effective towards infected person/group then there is less possibility to recover or die.” I am assuming that there is a lower possibility of recovering rather than dying.
• The structure of the paper largely conforms to PeerJ standards and is as expected for an interdisciplinary review.
• The review is timely, of broad and cross-disciplinary interest and within the scope of the journal.
• However, the field has been extensively reviewed (a quick PubMed search revealed several such papers: https://pubmed.ncbi.nlm.nih.gov/?term=artificial%20intelligence%20covid-19%20review). It is unclear what distinguishes this review from existing ones. This would be a good point to emphasize in the abstract and the introduction.
• The Introduction does not explicitly make it clear which audience the review is addressing. However, the implication is that this is a review meant to initiate AI researchers towards COVID-19 work.

Experimental design

• The content is within the aims and scope of the journal.
• While I cannot comment on the ethical standard of the work done for this review, I think on the technical side, the review could be carried our more systematically (see below).
• There is a great lack of detail in the description of the methods and I highlight two major points here. First, it is unclear what search terms and their combinations were used to query the preprint servers. Were any synonyms such as “machine learning” used? Second, what were the inclusion/exclusion criteria for the manual selection of papers?
• By using preprint servers as a primary resource to search for papers, I am concerned that the Survey Methodology is biased towards work that is not peer-reviewed or is undergoing peer review. This suggests a wide range of maturity of these articles and it is not clear whether this is accounted for when choosing and highlighting papers. The survey largely ignores the body of work published in peer-reviewed journals. At the very least, the survey would be more comprehensive if open access journal articles were included in this review. Alternatively, providing clear statistics on the fates of these articles and the process to eliminate works-in-progress can be considered.
• Sources are adequately cited.
• While the review is coherently organized into sections and sub-sections, the content in each of these is inconsistent with the title. For example, Section 7 is titled “Treatment of COVID-19” but its sub-sections largely focus on radiology images, which has more to do with diagnosis. On a related note, there is a much larger issue pervasive throughout the paper that I discuss below in my general comments.

Validity of the findings

• Since this is a review, there are no real conclusions to draw or goals to be met but the paper does a modest job in providing a broad overview of this topic.
• The review does a reasonable job in outlining future directions. However, each sub-section is sparse on details and does not come across as a well thought out roadmap for the field. I am also not convinced by the use of drones to combat COVID-19 as issues such as trust and safety will need to be addressed. A better approach would be to adopt a more speculative tone and condense this section to 2-3 broad sections that talk about ongoing work. This will help demonstrate feasibility.

Additional comments

• The most important concern that I have is that there is a recurring abuse of terminology when it comes to the COVID-19 aspects of the paper. For instance, prediction is a broader term that spans can be made at the patient level (the progression of the disease) or the population level (the spread of the disease). On the other hand, diagnosis specifically focuses on the patient and involves either assigning a COVID-positive or COVID-negative status or more deeper classification (e.g., severity). In this paper, these terms are used interchangeably and are confusing when trying to follow the organization of the review. Similarly, in line 175, it is debatable whether apps that collect responses to questionnaires are actually “predicting” COVID-19 status.
• The lack of mastery of terminology, particularly in molecular biology, comes through and adversely affects the quality of the review. For instance, in line 243, the RCSB Protein Data Bank contains protein structures (and not “gene structures” as mentioned here). Another example, is the following phrase in line 610: “cell-lines of existing medicines”. What does this mean?
• Are Tables 1 and 2 directly extracted from existing papers or were they generated for this review? If it is the latter, what data were used to populate these tables and how were these constructed?
• The paper is very inconsistent in when it chooses to highlight the performance of tools and when it merely reports the existence of a particular tool. What was the rationale for diving deeper for some papers?

---

## Round 0.2 · accepted · Accept

Based on my own reading and that of one of the original reviewers, I recommend acceptance of this timely literature review in its current form.

Reviewer 1 ·

Basic reporting

It meets the standards of a scientific journal.

Experimental design

It falls within the scope of the journal.

Validity of the findings

The methodology was chosen correctly, the results are valid.

Additional comments

All my comments have been taken into account, I recommend acceptance of this work.